# Transcriptome Analysis of *Picea crassifolia* in Response to Rust Infestation

**DOI:** 10.3390/jof10050313

**Published:** 2024-04-25

**Authors:** Hailan Li, Luchao Bai

**Affiliations:** College of Agriculture and Animal Husbandry, Qinghai University, Xining 810016, China; lihailan1119@foxmail.com

**Keywords:** *Picea crassifolia*, transcriptome analysis, rust infestation, stomata, wax layer, lignin

## Abstract

This study examines the relationship between needle age and rust resistance in *Picea crassifolia*, focusing on the needle morphology, including size, shape, and physiological traits. One-year-old spruce needles are more susceptible to rust, while two-year-old needles show effective resistance. Using RNA-seq on the Illumina HiSeq500 platform, we analyzed both healthy and diseased one-year-old needles (N and B), as well as healthy one-year-old and two-year-old needles (N and L). We applied a fold change (FC) threshold of ≥2 and a false discovery rate (FDR) of <0.01, alongside GO annotation and KEGG pathway enrichment, to identify differentially expressed genes (DEGs). In N vs. B, DEGs were significantly enriched in processes such as metabolism, cellular function, catalysis, binding, ribosomal function, plant-pathogen interactions, endoplasmic reticulum protein processing, and signal transduction, revealing a polygenic network regulating the rust response. Similarly, in N vs. L, electron microscopy highlighted morphological differences in the wax layers of needles, with subsequent transcriptome sequencing uncovering genes involved in the development of one-year-old and two-year-old needles. DEGs were primarily found in pathways related to cutin, suberin, wax biosynthesis, fatty acid metabolism, photosynthesis, and phenylalanine synthesis. Two-year-old needles displayed reduced stomatal density, higher lignin content, and a thicker wax layer compared to one-year-old needles. Validation of the RNA-seq data through RT-qPCR on 10 DEGs confirmed the consistency of gene expression trends, enhancing our understanding of *Picea crassifolia*’s genetic response to rust and supporting future research into its disease resistance.

## 1. Introduction

*Picea crassifolia* Kom., a native tree species in Qinghai Province, exhibits exceptional resilience to cold, drought, and saline-alkali conditions. It plays a crucial role in soil and water conservation efforts, contributing significantly to water retention [1]. The needle morphology of *Picea crassifolia*, including its size, shape, and physiological characteristics, stands as a pivotal factor enabling plants to withstand stress. However, within the same tree species, notable variations exist in ecological adaptability and disease resistance, especially in resistance to rust [2]. For instance, while needles of one-year-old spruce are susceptible to rust infection, those of two-year-old spruce demonstrate effective resistance against this disease. The onset of *Picea crassifolia* rust typically occurs from June to July. Initially, yellow spots appear on the needles of annual spruce, marked by needle-like dots identified as sporophores [3]. Subsequently, numerous orange, slightly raised oval or strip-shaped spore organs develop. As these rust spore organs mature, their capsules rupture, releasing yellow powdery rust spores [4]. By early August, the affected leaves turn grayish-yellow and gradually dry out. Spruce rust is a significant disease affecting spruce seedlings, young trees, and mature trees [5]. The disease damages terminal and lateral buds, severely impacting the quality of seedlings and the normal growth of plantations. In severe cases, it leads to widespread needle death and even branch dieback [6], resulting in substantial economic losses and reducing the trees’ ornamental value. Rust has the phenomenon of transferring to host, which greatly affects the prevention and control of disease [7], making effective management strategies critical [8].

Throughout their growth and development, plants are constantly exposed to microorganisms like bacteria, viruses, and fungi. Although plants lack a circulatory immune system, they have developed a variety of resistance mechanisms through evolution [9,10]. When plant cells detect an invading pathogen, they rapidly signal neighboring cells, initiating a series of defensive responses [11]. The primary defense is the cell wall, a physical barrier composed of complex polysaccharides coated with a wax layer that enhances its protective function. The synthesis of glycoproteins, cellulose, and lignin further strengthens the cell wall [12].

The secondary line of defense is chemical. Plants detect pathogens and insects through their secretions and other molecular features, which interact with plant cell surfaces. This interaction triggers plant signaling molecules, activating signal cascades that turn on defense and resistance genes [13]. In addition, the influx of calcium ions and the production of reactive oxygen species and nitrogen species activate mitogen-activated protein kinase (MAPK). MAPKs then trigger transcription factors that influence the expression of pathogenesis-related genes (PR), stimulate the production of ethylene, jasmonic acid (JA), and salicylic acid (SA), enhance the plant cell wall, and induce the synthesis of antibacterial compounds [14].

High-throughput sequencing technology has found wide application in biological research [15]. It enables precise analysis of gene differential expression and functional annotation, thereby deepening our comprehension of fungal pathogenic mechanisms [16]. This innovative technology has been utilized by researchers to enhance the resistance abilities among plants. Leveraging transcriptome technology, researchers have examined the variance in gene expression among different pathogenic wheat leaf rust summer spores, uncovering 12 genes that may play a pivotal role in wheat leaf growth and rust disease pathogenicity [17]. Moreover, the significance of leaf surface characteristics in plant defense is also well established. Research has revealed that these characteristics, governed by multiple genes, influence stomata formation and the development of the wax layer [18]. A comprehensive understanding of these traits and associated gene expressions offers valuable insights into plant defense mechanisms against diseases [19]. For example, research on Populus euphratica revealed age-related differences in stress resistance and photosynthetic traits [20]. Stomatal density is used as an initial measure for disease resistance, with leaf thickness and spongy tissue thickness serving as additional indicators. Generally, a higher stomatal density correlates with lower disease resistance [21]. Currently, there is a scarcity of research on Picea crassifolia rust at the transcriptome level. Therefore, our study aimed to fill this gap by conducting transcriptome sequencing on healthy one-year-old (N), healthy two-year-old (L), and diseased one-year-old (B) needles of Picea crassifolia to identify differentially expressed genes (DEGs). This study contributes significantly to understanding the role of these genes in the Picea crassifolia response to rust infection, laying the groundwork for future genomic research on the tree’s disease resistance or susceptibility.

## 2. Materials and Methods

### 2.1. Sample Collection

On 1 July 2022, plant samples were gathered from the Huzhu Beishan forest, situated in Huzhu County, Xining City, Qinghai Province, China (at an altitude of 1738 m; 107°36′ E, 26°18′ N). These samples were collected under natural conditions with an annual average temperature of 4.6 °C and precipitation ranging from 400–600 mm. Healthy one-year-old needles (N) (Figure 1A), healthy two-year-old needles (L) (Figure 1B), and diseased one-year-old needles (B) (Figure 1C) were collected from each plant as sequencing samples. Subsequently, the samples were frozen in liquid nitrogen and transported to biological companies for sequencing analysis.

### 2.2. Sample Preparation and Microscopic Observation

The samples were initially placed under a pose microscope (Nikon SMZ1500-FIL, Nikon Corporation, Tokyo, Japan) for photographic observation. The process involved fixing the spruce needles with 4% glutaraldehyde, followed by rinsing with a phosphate buffer. Subsequent steps included gradient dehydration with a series of ethanol concentrations, replacement with isoamyl acetate, freeze-drying, mounting on stubs, and coating with gold. The prepared specimens were then examined and imaged using an electron microscope (JSM-7900F, Japan Electronics Corporation, Tokyo, Japan) [22].

### 2.3. RNA Isolation

RNA was isolated following the protocols recommended by the kit manufacturers. The nucleic acids were extracted using a combination of kits (DP411, CLB+ Adlai RN40, CTAB+ Adelai RN40, DP762-T1C) and TRIzol reagent. The quality and concentration of the extracted RNA were assessed using a Nanodrop2000 spectrophotometer (Thermo Finsher Scientific, Waltham, MA, USA). The RNA integrity was further evaluated using an Agilent 2100 bioanalyzer in conjunction with a Lab Chip GX system (PerkinElmer, Waltham, MA, USA). For downstream applications, half of each RNA sample was directed to cDNA library construction and sequenced on the Illumina HiSeq6000 platforms owned by Biomarker Technologies (San Diego, CA, USA). The remaining half was reserved for verification via quantitative RT-PCR (qRT-PCR).

### 2.4. Data Processing

To ensure a smooth data analysis process, it is imperative to obtain a sufficient number of high-quality reads to ensure the accuracy of sequence assembly and subsequent bioinformatics analysis. In this study, FastQC (version 0.11.9) [23] was employed to evaluate the read quality. The connectors in the original sequencing data (Raw Date) and low-quality reads were filtered to obtain high-quality sequences (clean data) for further analysis. Subsequently, GC content and Q30 were used as the quality standard of sequencing data [24]. Trinity v2.13.2 was used for transcriptome assembly and DEseq2 v1.22.2 was employed for differential expression analysis. Differential genes were then screened by differential multiple |log2Fold Change| ≥ 2 and an error detection rate (FDR) < 0.01. ClusterProfiler v4.6.0 was utilized to analyze the enrichment of DEG in the KEGG pathway [25].

### 2.5. RT-qPCR Validation of Differential Gene Expression

To verify the reliability of transcriptome data, we randomly selected 10 genes for qRT-PCR expression detection, with the *SKI2* gene serving as the control (Table 1). The quantitative PCR reagent was ChamQ SYBR Color qPCRMaster Mix (2×), Nanjing Nuoweizan Biotechnology Co., Ltd. (Nanjing, China), (Table 2) and the amplification program of the Bori LineGene9600plus fluorescence quantitative PCR instrument was 95 °C 3 min, 35 cycles (95 °C, 30 °C, 30 min, 72 °C, 40 s), and the relative expression was calculated by 2^−ΔCt^. The PCR reaction steps and the test materials and instruments used are shown in the table below (Table 3 and Table 4).

## 3. Results and Analysis

### 3.1. Pathogen Identification

Through comprehensive morphological analysis, the pathogen causing rust disease in *Picea crassifolia* was identified as belonging to the Melampsoraceae family. This pathogen falls within the Basidiomycete class, part of the Pucciniomycotina subphylum and the Pucciniales order. Characterized by a cyclical life cycle, the pathogen produces rust spores on *Picea crassifolia* needles (Figure 2A,B), with teliospores found on the undersides of Rhododendron qilianense leaves (Figure 2C,D). These rust spores are light yellow, predominantly round or oval, and range in size from 26 to 34 μm in length and 17 to 24 μm in width. Notably, the surfaces of these spores are densely covered with small warts. Teliospore clusters, which are head-shaped and enveloped in a wax-like protective layer, exhibit an orange–red color. The base of these clusters, known as basal fascicles, is light yellow and measures approximately 0.2 to 0.5 mm in length [26].

### 3.2. Differences in Stomatal Morphology of Spruce Needles

Under a stereomicroscope, we examined the shape and distribution of stomata on the spruce needles. The needles, quadrangular in shape, feature stomata aligned in straight lines on the epidermis (Figure 3). Each surface has 2–6 white lines of stomata, alternating with the needle edges. Notably, the two-year-old needles have fewer stomata compared to the one-year-old needles. Further examination under a scanning electron microscope revealed notable differences in stomatal size. The one-year-old needles exhibited larger stomatal openings, whereas the two-year-old needles had smaller openings with a distinct overlay covering them, as shown in Figure 4.

### 3.3. Morphological Differences in the Wax Layer of One-Year-Old and Two-Year-Old Spruce Needles

Using a higher magnification on the scanning electron microscope, we observed the wax layer morphology on the spruce needles. The stomata on the needle surface are recessed, with noticeable variations within the wax layer. On the one-year-old needles, the wax layer appears filamentous and thinner, whereas on the two-year-old needles, it is distributed more uniformly in patches around the stomata and is noticeably thicker. These findings are illustrated in Figure 5.

### 3.4. Sequencing Data and Quality Assessment

We conducted transcriptome sequencing on healthy one-year-old spruce needles (N1, N2, N3), healthy two-year-old needles (L1, L2, L3), and diseased one-year-old needles (B1, B2, B3). The sequencing yielded nine sets of high-quality data (clean reads). The GC content in these data sets was above 43.83%, and the quality score (Q30) exceeded 93.01% (Table 5). These results indicate that the data quality is excellent, making it suitable for further analysis.

### 3.5. Screening of Differentially Expressed Genes

As shown in Figure 6, we identified 15,408 DEGs in the comparison between healthy and diseased one-year-old needles (N vs. B), which included 10,296 upregulated genes and 5112 downregulated genes. In the comparison between healthy one-year-old and two-year-old needles (N vs. L), we found 3269 DEGs, comprising 1469 upregulated and 1800 downregulated genes Figure 6.

### 3.6. Analysis of Differentially Expressed Gene Functions Using GO Annotation

In our gene ontology (GO) analysis, the DEGs were categorized into three main groups: biological process, cellular component, and molecular function, as illustrated in Figure 7. In both N vs. B Figure 7 and N vs. L Figure 7 comparisons, DEGs were mainly involved in metabolic and cellular processes, activities specific to single organisms, biological regulation, response to stimulus, and cell structure and development under the biological process category. Within the cellular component category, DEGs were primarily found in cells, cell parts, organelles, and membrane-related areas. For molecular function, the DEGs showed significant involvement in binding and catalytic activities.

### 3.7. KEGG Pathway Analysis of Differentially Expressed Genes

The 15,408 DEGs identified in the N vs. B comparison were further analyzed using KEGG pathway annotation. The most prominent pathways included plant-pathogen interactions, carbon metabolism, amino acid biosynthesis, endoplasmic reticulum protein processing, MAPK signaling in plants, plant hormone signaling, and starch and sucrose metabolism, Figure 8. In the N vs. L comparison, the 3269 DEGs also underwent KEGG pathway annotation. The key pathways enriched in this comparison were similar, including plant-pathogen interactions, MAPK signaling in plants, plant hormone signaling, starch and sucrose metabolism, amino acid biosynthesis, carbon metabolism, and endoplasmic reticulum protein processing, Figure 8.

### 3.8. Validation of Differentially Expressed Genes by RT-qPCR

To ensure the reliability of our RNA-seq data, we selected 10 differentially expressed genes related to the disease for RT-qPCR expression detection. The differentially up-regulated genes include 044105 (*MEKK1*, mitogen-activated protein kinase 1), 046243 (*MEKK1*, mitogen-activated protein kinase 1), 137891 (*GH3*, auxin-responsive GH3 family), 150025 (*NDK*, *NME*, nucleoside diphosphate kinase), and 136462 (*DDC*, *TDC*, aromatic-L-amino acid/L-tryptophan decarboxylase). The differentially down-regulated genes encompass 047103 (*WRKY33*, *WRKY* transcription factor 33), 114172 (*EFR*, *LRR* receptor-like serine/threonine protein kinase EFR), 151400 (*MPK6*, mitogen-activated protein kinase 6), 261811 (*TIR1*, transport inhibitor response 1), and 044230 (*phhA*, *PAH*, phenylalanine 4-hydroxylase). The internal reference is *SKI2*. The results showed congruent gene expression patterns between RT-qPCR and RNA-seq analyses (Figure 9), affirming the reliability of the transcriptome sequencing data. This validates their suitability for subsequent analyses of differentially expressed genes.

## 4. Discussion

At present, there is limited research at the transcriptome level regarding *Picea crassifolia* rust. Therefore, in this study, we sequenced the needles of different leaf ages in *Picea crassifolia*, initially uncovering the general expression patterns at the transcriptome level and identifying pertinent DEGs for subsequent metabolic pathway analysis. This study significantly contributes to understanding the role of these genes in *Picea crassifolia*’s response to rust infection, laying the groundwork for future genomic research on the tree’s disease resistance or susceptibility.

In the comparative analysis of N vs. B samples, the study observed that plant cells, upon detecting an invading pathogen, rapidly propagate this signal to adjacent cells, triggering a sequence of defense mechanisms [24,25,26,27]. These include the occurrence of an oxidative burst, characterized by a swift accumulation of reactive oxygen species (ROS) [28]. Signaling molecules such as calcium (Ca^2+^), hydrogen peroxide (H_2_O_2_), and nitric oxide (NO) play pivotal roles in the hypersensitive response (HR) signaling amid pathogen attacks [28]. H_2_O_2_, in low concentrations, activates cellular antioxidant systems, yet in higher concentrations, it disrupts these systems, leading to oxidative damage and programmed cell death [29]. This study’s findings align with research indicating an inverse relationship between H_2_O_2_ levels and resistance to black spot disease in roses, suggesting higher H_2_O_2_ accumulation in diseased plants [30]. Calcium (Ca^2+^) plays a key role as an intracellular messenger in plants, participating in a range of biochemical reactions and signal transduction processes [31]. It works alongside nitric oxide (NO) in regulating various aspects of plant growth and health, including cell proliferation, metabolism, and cell death. Both Ca^2+^ and NO are important in managing the plant’s antioxidant defenses, especially under stress. Notably, NO is known to trigger the expression of genes responsible for antioxidants and boost the activity of antioxidant enzymes [32]. In the present work, the activation of the MAPK plant signaling pathway is observed, involving components such as *MKK4/5*, *MPK3/6*, and the *WRKY29* transcription factors. This activation is associated with increased production of H_2_O_2_. Concurrently, there is a decrease in the activity of nitric oxide synthase (NOS) within the plant-pathogen interaction pathway, leading to reduced synthesis of NO and a lower expression of genes related to antioxidants. These changes suggest that the rise in H_2_O_2_ levels, coupled with diminished NO synthesis, may impair the plant’s ability to resist disease, potentially making it more susceptible to infection.

The fundamental signaling pathways of plant defense responses are influenced by various hormones, such as jasmonic acid (JA) and salicylic acid (SA). These two opposing pathways, JA and SA, play critical roles in protecting plants from pathogen invasion [33]. They enable plants to react to pathogen-induced stress through signal transduction processes involving pathogens like Magnaporthe oryzae [34,35]. Research has demonstrated that during the infection process, Magnaporthe oryzae secretes an effector protein that interferes with the host’s JA-induced defense mechanisms [36]. Furthermore, it exploits the antagonistic interaction between the JA and SA signaling pathways to suppress the SA-mediated resistance, enhancing its pathogenic effectiveness. Some researchers have found that the JA produced by pathogens such as L. theobromae during host infection suppresses the synthesis of the host’s SA, thereby inhibiting systemic acquired resistance (SAR) due to the reduced SA levels [37].

It is widely recognized that the SA pathway plays a crucial role in mitigating the impact of biotrophic and hemibiotrophic pathogens [38]. We hypothesize that L. theobromae may suppress SA synthesis by releasing substantial amounts of JA, subsequently hindering the host’s induced resistance via the SA signaling pathway, thereby facilitating its pathogenic capabilities [39,40,41]. In this study, the PR1 (pathogenesis-related protein 1) gene did not appear active in the plant signaling pathway, plant-pathogen interactions, or plant hormone signaling induction pathways. Genes associated with SA, such as TGA and NPR1, were observed to be down-regulated. We speculate that rust pathogens may infect plants by emitting large quantities of JA, which then inhibits SA synthesis and contributes to the development of the disease.

Furthermore, we observed a noteworthy suppression in the expression of genes linked to photosynthesis. Chloroplasts, pivotal to this process, contain two key photosynthetic systems, namely, Photosystem I (PSI) and Photosystem II (PSII) [42]. PSI plays a role in reducing NADP, whereas PSII is critical for the photolysis of water and releasing oxygen [43,44]. The combined activities of PSI and PSII enhance the efficiency of photosynthesis within chloroplasts. They also contribute significantly to the production of ROS in plants. The reduced expression of photosynthesis-related genes leads to a significant decrease in the plant’s ability to assimilate carbon dioxide (CO_2_). To compensate, plants increase the production of H_2_O_2_ through photorespiration and peroxisome activities, a response aimed at improving their growth conditions under stress [45,46]. In this study, genes involved in PSI and PSII, as well as those in cytochrome, light and electron transfer chains, and carbon fixation in photosynthesis, were found to be downregulated in diseased *Picea crassifolia* plants. This downregulation impacted the plants’ light capture capabilities and the efficiency of electron transfers in PSI and PSII, leading to reduced photosynthetic efficiency. As a result, there was notable leaf chlorosis and necrosis in photosynthetic tissues, culminating in a diminished capacity for carbon assimilation [47,48,49,50,51].

In the investigation of the N vs. L group, we observed that the primary defense against pathogens in plants is situated at their surface. This defense includes structures like wax layers on leaves and the cuticle covering epidermal cells, which provide a barrier to pathogen invasion [52,53]. Among these defenses, stomata, the tiny openings on leaf surfaces, are particularly significant. They serve as pathways for gas and water exchange between the plant and its environment and are essential in processes like carbon assimilation, respiration, and transpiration [54]. Research in this area has often focused on agricultural crops, with less emphasis on tree species. Previous studies by researchers such as Xie Wenhua, Gu Zhenfang, Zhao Zhenling, and Chai Chenbo have demonstrated a clear link between stomatal density and disease resistance. They found that plants with a lower stomatal density generally exhibit stronger resistance to diseases like powdery mildew and black spots, whereas those with a higher stomatal density tend to be more susceptible [55,56,57,58]. This suggests that larger stomatal openings may increase the likelihood of pathogen entry, thereby reducing the plant’s overall disease resistance [59].

Keys to the development of stomata are transcription factors such as *SPCH* (SPEECHLESS), *MUTE*, and *FAMA* [60]. *SPCH* initiates the development process by driving cell divisions, while MUTE and FAMA contribute to the formation of guard cells, which are crucial in regulating stomatal patterns [61,62]. In this study, we noted an up regulation in the gene expression of *SPCH*, *MUTE*, *FAMA*, and certain enzymes involved in stomatal formation in the needles of two-year-old conifers. Interestingly, these needles exhibited a lower stomatal density compared to those of the one-year-old conifers, aligning with findings from previous research.

Plant cuticle wax is critical for adapting to external challenges like drought and environmental stress [63]. Research into the development of rice leaf cuticle wax and transcription factors involved in wax synthesis has revealed that plant varieties resistant to diseases such as powdery mildew in bitter gourd generally exhibit a higher wax content than their susceptible counterparts [64,65]. Additionally, studies by Wang and Feng have shown that tree species with greater disease resistance tend to have more wax on their leaves, and the removal of this wax layer increases their vulnerability to diseases [66,67]. The process of synthesizing and transporting plant cuticle wax is complex, involving numerous genes [68]. The components of this wax include very long-chain fatty acids, their derivatives, alkanes, aldehydes, alcohols, ketones, terpenes, and various small-molecule secondary metabolites [69,70]. In this study, we observed that the wax layer on the needles of two-year-old conifers was considerably thicker than on the one-year-old needles. This observation was supported by the upregulated expression of enzymes involved in wax synthesis. These enzymes were active in the pathways related to “cuticle, suberization, and wax biosynthesis”, “fatty acid biosynthesis”, “unsaturated fatty acid biosynthesis”, and “photosynthesis” in the needles of the second-year spruce. This evidence suggests that the wax layer plays a significant role in the response of spruce needles to rust infection, possibly contributing to their defense mechanism.

Lignin serves as a key structural component in plant cell walls, with its increased presence leading to stronger cell walls, reduced tissue damage, and improved resistance to diseases [71]. The biosynthesis of lignin involves a series of enzymes, such as *COMT* (caffeic acid-O-methyltransferase), *CCoAOMT* (caffeoyl-CoA O-methyltransferase), *F5H* (ferulate 5-hydroxylase), phenylalanine ammonia-lyase (*PAL*), cinnamate 4-hydroxylase (*C4H*), and 4-coumarate-CoA ligase (*4CL*). These enzymes play a significant role in determining the total lignin content in plants [72,73,74]. *PAL*, a critical enzyme in the phenylalanine pathway, links the coumaric acid pathway to the phenylpropanoid pathway, significantly influencing lignin synthesis [75,76]. Similarly, *4CL* acts at the end of the phenylpropanoid pathway, facilitating the transition to specific lignin synthesis [77]. It encodes for 4-coumarate:CoA ligase, a key enzyme at the end of the phenylpropanoid pathway. In addition, enzymes like CAT (catalase) and POD (peroxidase) are significant in the lignin synthesis process. They contribute to the polymerization of lignin precursors, promote cell wall lignification, and aid in forming protective cork cells near wounds, thereby enhancing the plant’s resistance to diseases [78,79,80].

Moreover, we observed an increased expression of *CCoAOMT*, *PAL*, *4CL*, 2-hydroxyacyl-CoA lyase, 2,4-dienoyl-CoA isomerase, hydroxymethylglutaryl-CoA lyase, CAT, and POD in the needles of perennial conifers, with a corresponding decrease in the expression of *COMT*. This pattern of enzyme expression suggests an upsurge in lignin production in the second-year needles. These enzymes are integral to lignin synthesis, participating in the conversion of lignin precursors, enhancing the lignification of cell walls, and aiding in the formation of protective cork cells near wounds. The increased lignin content in these second-year needles is likely a factor contributing to their enhanced structural integrity and disease resistance.

## 5. Conclusions

This study conducted transcriptome sequencing on both healthy and diseased samples of *Picea crassifolia* to identify their DEGs. To confirm the accuracy of the RNA-seq data, RT-qPCR expression analysis was performed on 10 DEGs. The findings affirm that the gene expression trends observed in RT-qPCR are consistent with those from RNA-seq.

In the comparison of healthy (N) and diseased (B) samples, the analysis revealed a significant enrichment of DEGs in critical pathways, including photosynthesis, plant signaling, interactions with pathogens, hormone signaling in plants, and phenylalanine biosynthesis. Notable genes within these pathways, such as *PR1*, *MKK4/5*, *MPK3/6*, *WRKY29*, *NOS*, *TGA*, and *NPR1* were identified as key players in the plant’s response to rust infection.

Focusing on the comparison between healthy one-year-old and two-year-old needles (N vs. L), we used electron microscopy to observe the morphological differences in stomatal and wax layers. The second-year needles showed notable traits of reduced stomatal density, increased lignin content, and a thicker wax layer compared to the first-year needles. Further analysis through GO and KEGG enrichment revealed that genes involved in stomatal development (e.g., *SPCH*, *MUTE*, *FAMA*) and enzymes critical for stomatal formation were more active in these older needles. The study also highlighted the upregulation of enzymes related to wax syntheses, such as fatty acid omega-hydroxylase, fatty acid omega-hydroxydehydrogenase, catalase, aldehyde decarbonylase, acetyl-CoA acyltransferase, and acetyl-CoA acyltransferase 1. Additionally, the increased expression of genes involved in lignin production, including *CCoAOMT*, *PAL*, *4CL,2-hydroxyacyl-CoAlyase*, *2,4-dienoyl-CoA* isomerase, hydroxymethylglutaryl-CoA lyase, CAT, and POD in the perennial needles, suggests a strengthening of the needle structure and an enhancement in disease resistance.

This comprehensive study enhances our understanding of the genetic reactions of *Picea crassifolia* to rust infection, laying the groundwork for future investigations in plant pathology and genetics. It especially contributes to the knowledge regarding disease resistance mechanisms in conifer species.

## Figures and Tables

**Figure 1 jof-10-00313-f001:**
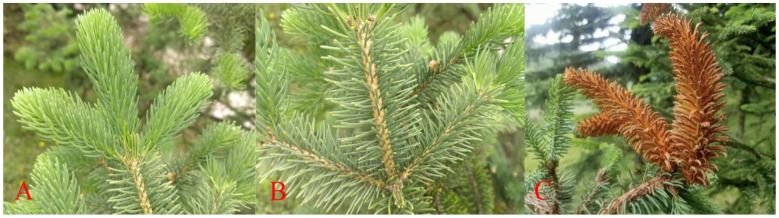
*Picea crassifolia* Kom: (**A**) Healthy one-year-old needles, (**B**) Healthy two-year-old needles, and (**C**) Diseased one-year-old needles.

**Figure 2 jof-10-00313-f002:**
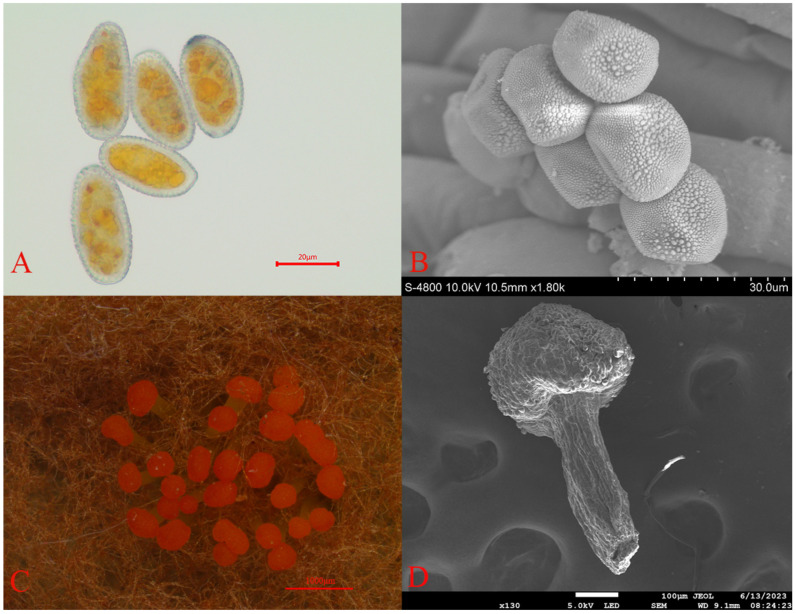
Micrographs of the rust spores: (**A**) rust spores (Optical microscope (FL-20)); (**B**) rust spores (electron microscope (JSM-7900F)); (**C**) Teliospore pile (Somatotype microscope (SMZ1500-FIL)); and (**D**) Teliospore pile (electron microscope (JSM-7900F)). Scale bars: (**A**) = 20 μm; (**B**) = 30 μm; (**C**) =1000 μm; and (**D**) =100 μm.

**Figure 3 jof-10-00313-f003:**
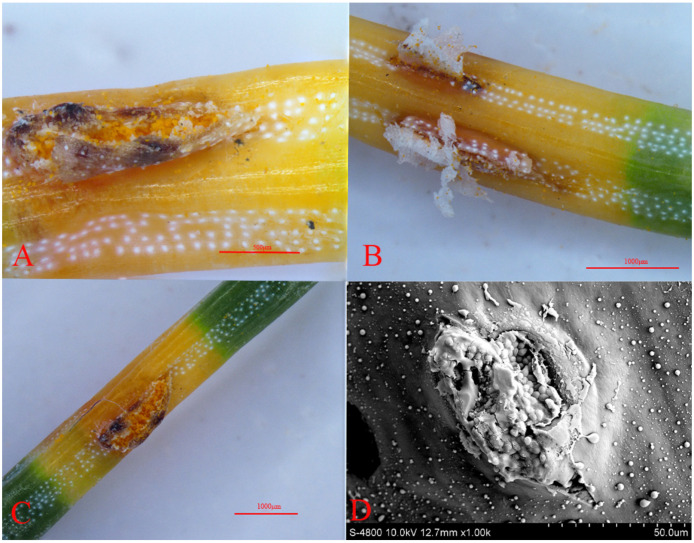
Stomata of spruce: (**A**–**C**) (Somatotype microscope (SMZ1500-FIL)); and (**D**) electron microscope image (JSM-7900F). Scale bars: (**A**) = 500 μm; (**B**,**C**) = 1000 μm; and (**D**) = 50 μm.

**Figure 4 jof-10-00313-f004:**
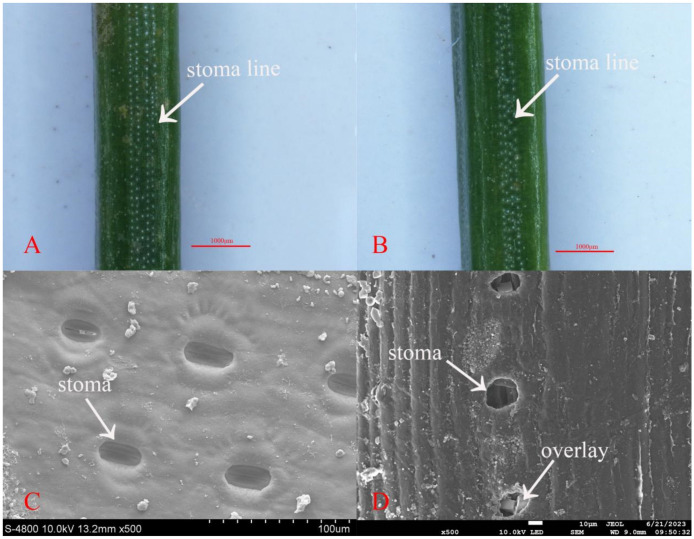
Scanning electron microscopic images of stomata on spruce needles: (**A**,**C**) One-year-old spruce; (**B**,**D**) Two-year-old spruce. (**A**,**B**) Somatotype microscope (SMZ1500-FIL); (**C**) electron microscope (JEOL, JSM–6360LV); and (**D**) electron microscope (JSM-7900F). Scale bars: (**A**,**B**) = 1000 μm; (**C**) = 100 μm; and (**D**) = 10 μm.

**Figure 5 jof-10-00313-f005:**
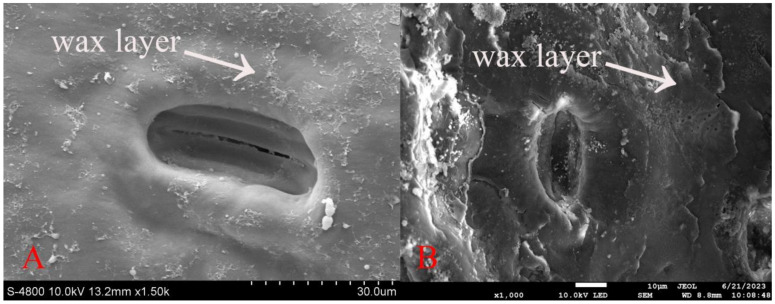
Differences in wax layer morphology on spruce needles: (**A**) One-year-old spruce (electron microscope (JEOL, JSM–6360LV)); and (**B**) Two-year-old spruce (electron microscope (JSM-7900F)). Scale bars: (**A**) = 30 μm; and (**B**) = 10 μm.

**Figure 6 jof-10-00313-f006:**
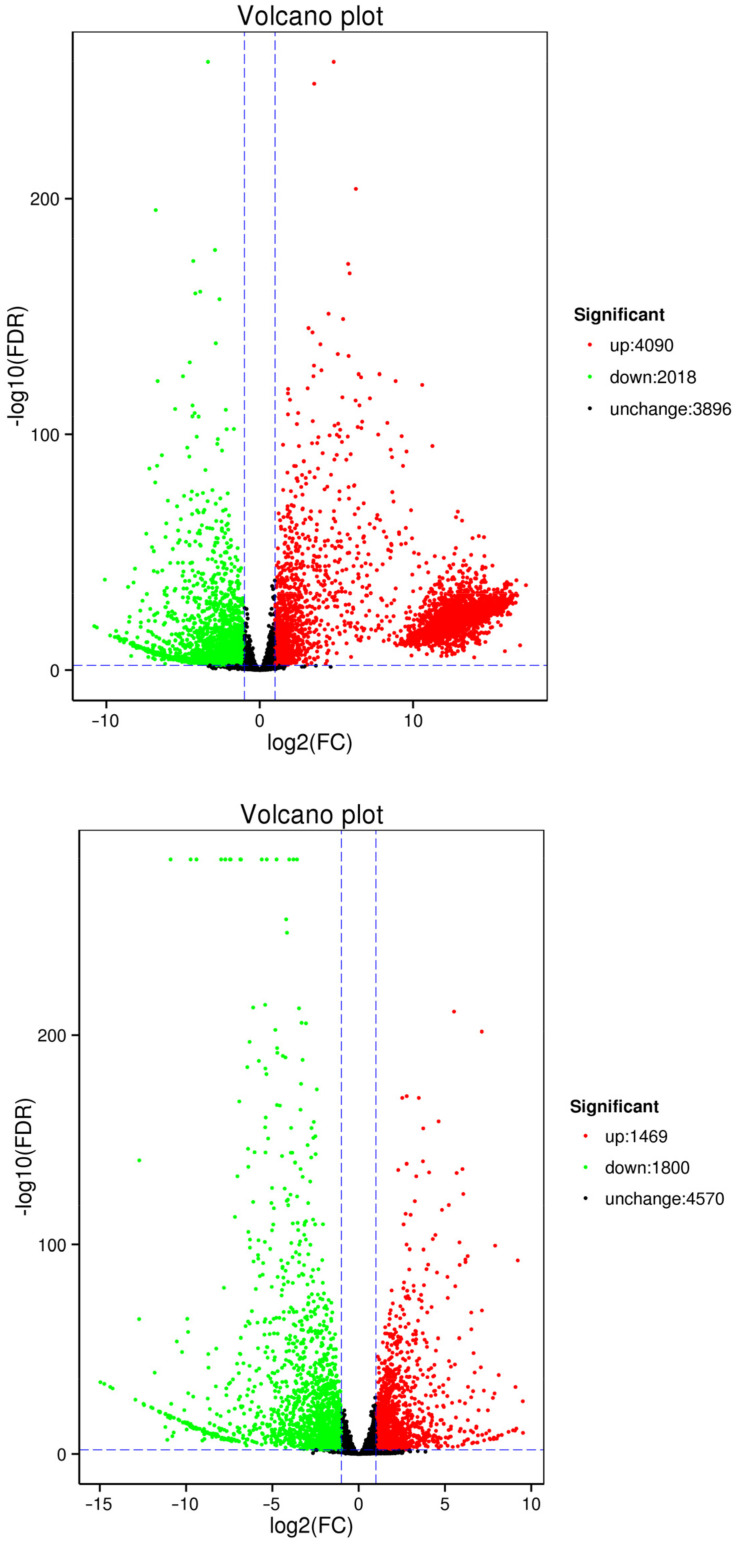
Volcano plots illustrating differentially expressed genes: Comparison of N vs. B (**up**) and N vs. L (**down**).

**Figure 7 jof-10-00313-f007:**
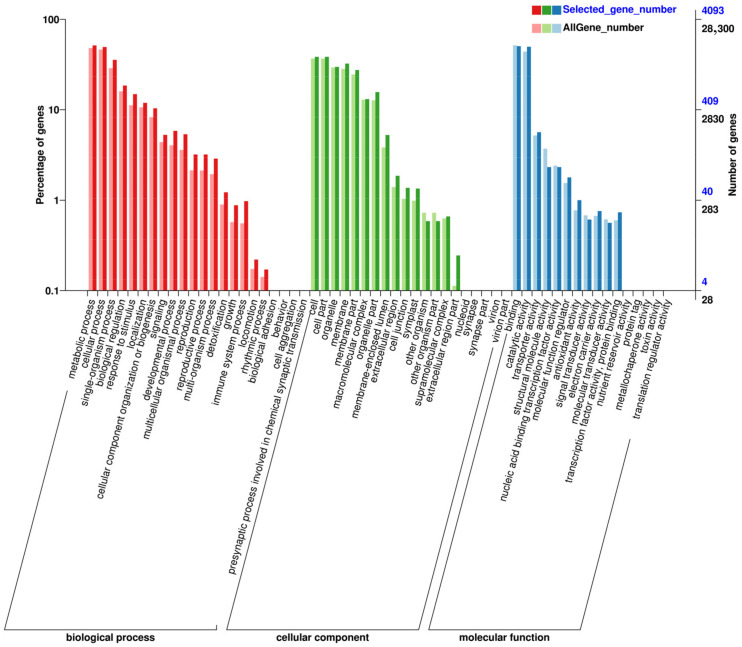
GO annotation analysis of differentially expressed genes: Comparison of N vs. B (**up**) and N vs. L (**down**).

**Figure 8 jof-10-00313-f008:**
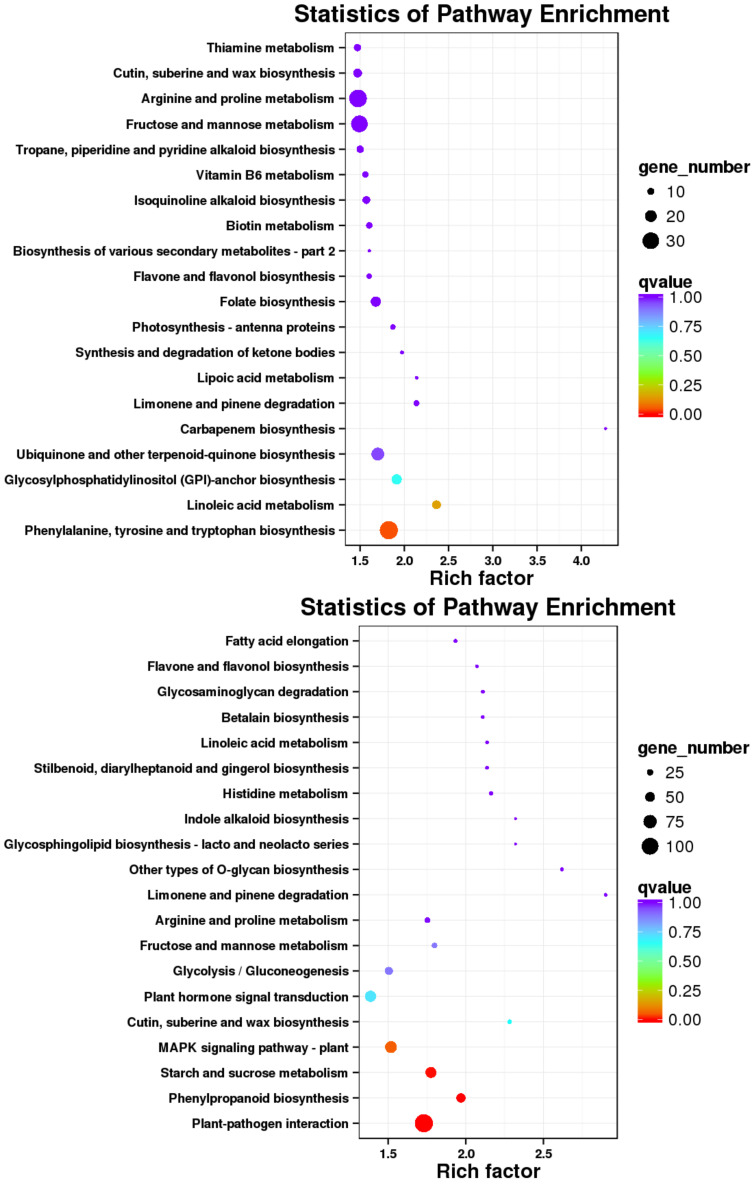
KEGG pathway enrichment analysis for DEGs: N vs. B (**up**) and N vs. L (**down**).

**Figure 9 jof-10-00313-f009:**
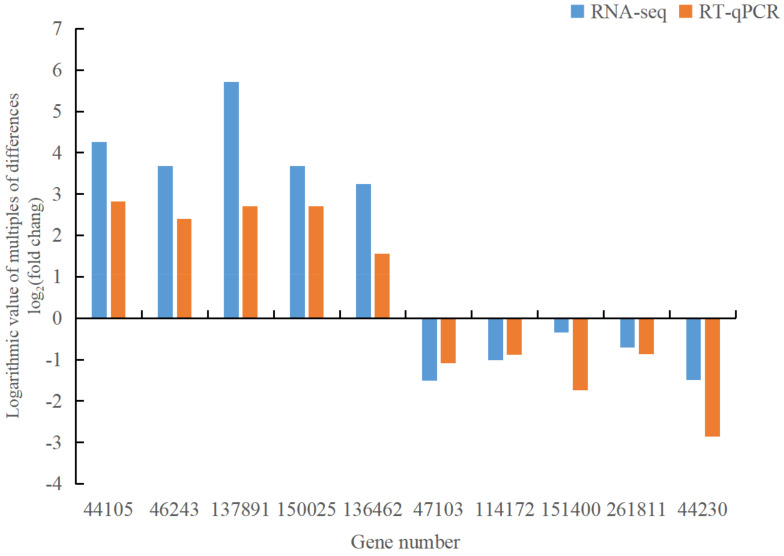
RT-qPCR validation results of 10 differentially expressed genes.

**Table 1 jof-10-00313-t001:** List of primers used for RT-qPCR analysis.

Gene Number	Primer Name	Sequences (5′ to 3′)	Clip Size
137891	s137891-F	F: GAAATTAGCTCCCGTGTTTC	176
	s137891-R	R: GCTCTCTACAGGAAAATCAC	
46243	s046243-F	F: TATGTTCAACCCGTCAAATC	183
	s046243-R	R: CAGTGGATGATCCAGTTGAG	
150025	s150025-F	F: TAGAAGAAAGAGGACCAGGG	186
	s150025-R	R: CTGCAGCTAGTACCTTTCCC	
44105	s044105-F	F: CTTCTCTTCCGCTCATTAAC	198
	s044105-R	R: CTGTGGATTGCTCTGATACC	
136462	s136462-F	F: TGCTGCTGCTACGAGACTTC	191
	s136462-R	R: TGTTACAGTTGGTCCCAACG	
261811	x261811-F	F: ATGACAGTCTTAAACAGCGG	191
	x261811-R	R: AGGACTGTTCGATTGATCTC	
114172	x114172-F	F: TTAGTAACACAAAGAAGCCG	179
	x114172-R	R: GAGGATGCCAAATTAGCTTC	
151400	x151400-F	F: ATTTCGATTCTCTTCTGGTC	197
	x151400-R	R: GGATGTTATCCATGGGATTC	
47103	x047103-F	F: TAATACGCCCTTCCTCTATC	182
	x047103-R	R: CAACGCGACAGATTAAAAAG	
44230	x044230-F	F: CATTAACATACACAGCTCCC	183
	x044230-R	R: TAGACCAGTTGTGGGTATGC	
*SKI2*	*SKI2*-F	F: CACGCCTTCACAGCAATCAG	291
	*SKI2*-R	R: TCCTGTAGCCCTCTTCCATCA	

**Table 2 jof-10-00313-t002:** Preparation of the main mixture.

Reaction Component	Concentration	Volume (µL)
ChamQ SYBR Color qPCR Master Mix	2×	10
Primer F	5 µM	0.4
Primer R	5 µM	0.4
Template (DNA)		2
ddH2O		7.2
Total		20 µL

**Table 3 jof-10-00313-t003:** The main experimental instruments used in this experiment.

Number	Instrument Name	Model	Manufacturer
1	PCR instrument	MG96+	Hangzhou Langji Scientific Instrument Co., Ltd., Hangzhou, China
2	Double stable time electrophoresis instrument	dyy-6c	Beijing Junyi Oriental Electrophoresis Equipment Co., Ltd., Beijing, China
3	Hole centrifuge	H1650	Hunan Changsha Xiangyi Centrifuge Instrument Co., Ltd., Changsha, China
4	Precision pipette	a suit of	Eppendorf Co., Hamburg, Germany
5	Vortex mixer	MixMax	Hefei Ebenson Scientific Instrument Co., Ltd., Hefei, China
6	Gel imager	JY04S-3C	Beijing Junyi Oriental Electrophoresis Equipment Co., Ltd., Beijing, China
7	Fluorescence quantitative PCR instrument	9600plus	Hangzhou Bori Technology Co., Ltd., Hangzhou, China
8	Ultra-trace visible light ultraviolet spectrophotometer	ND5000	Beijing Baitech Biotechnology Co., Ltd., Beijing, China

**Table 4 jof-10-00313-t004:** The main experimental reagents and consumables used in this experiment are shown in the table.

Number	Name of Reagent or Consumable Material	Model/Type	Supplier
1	2X ChamQ SYBR, COLOR qPCR Master Mix	Q411-02/03	Nanjing Nuoweizan Biotechnology Co., Ltd., Nanjing, China
2	PCR primer	PAGE purification	Shenggong Bioengineering Co., Ltd., Shanghai, China
3	Eight even tube	PCR-0208-C	Aisijin Biotechnology (Hangzhou) Co., Ltd., Hangzhou, China
4	DL2000	DL2501	Shanghai Jierui Biological Engineering Co., Ltd., Shanghai, China
5	EX Taq enzyme	RR001	Baori Doctor Material Technology (Beijing) Co., Ltd. (TaKaRa), Beijing, China
6	dNTP	BK6501A	Baori Doctor Material Technology (Beijing) Co., Ltd. (TaKaRa), Beijing, China
7	10 × EX Taq Buffer	AB5401A	Baori Doctor Material Technology (Beijing) Co., Ltd. (TaKaRa), Beijing, China
8	Primer		Shenggong Bioengineering (Shanghai) Co., Ltd., Shanghai, China
9	HiScript Q RT SuperMix for qPCR (+gDNA wiper)	R123-01	Nanjing Nuoweizan Biotechnology Co., Ltd., Nanjing, China
10	Soil RNA Kit	R6825-02	Omega Bio-Tek, Norcross, GA, USA
11	OminiPlant RNA, Kit(DNase I)	CW2598S	Beijing Kangwei Century Biotechnology Co., Ltd., Beijing, China
12	E.Z.N.A.^®^ HP Tota RNA, Kit	R6812-01	Omega Bio-Tek, Norcross, GA, USA

**Table 5 jof-10-00313-t005:** Statistics of sequencing data quality for each sample.

Brochure	Read Number	Base Number	GC (%)	%≥Q30
N1	27,757,413	8,310,361,054	44.91%	94.48%
N2	28,723,680	8,599,171,640	43.83%	94.23%
N3	26,402,330	7,901,675,174	44.58%	94.72%
L1	27,757,413	8,310,361,054	44.91%	93.26%
L2	28,723,680	8,599,171,640	43.83%	93.38%
L3	26,402,330	7,901,675,174	44.58%	93.01%
B1	20,530,290	6,146,385,256	47.86%	94.46%
B2	21,433,171	6,417,397,574	47.80%	93.80%
B3	21,080,933	6,311,108,490	48.67%	93.90%

## Data Availability

Data are contained within the article.

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
