# Peer review of "Transcriptome Analysis of Picea crassifolia in Response to Rust Infestation"

_jof, 2024, doi:10.3390/jof10050313_

Round 1
Reviewer 1 Report
The article sent by the authors is very interesting, deals with such a relevant topic as studying the response of spruce to infection by rust fungi, uses various modern methods such as transcriptome analysis and electron microscopy, and will be useful in phytopathology, especially as similar data are accumulating for other plants. The authors have done the work at a fairly high technical and scientific level. There are a few comments to improve the manuscript.
1. This is a mostly formal remark, but references in the text should be arranged in accordance with the requirements of the journal, i.e. their numbers should be written in bold coloured font in square brackets, it is better to clarify this on the site. Now there are only numbers in the text, and this should be corrected.
2. It is also a formal point, but the names of species and genera should always be written in italics. It is necessary to make amends to “Phytophthora infestans” (line 40), “Populus euphratica” (line 52), “Pinus massoniana” (line 63), “Picea cracifolia” (line 83 and line 99, line 390) and possibly something else. Please check the manuscript again.
3. Similarly, when referring to genes rather than proteins, they should be written in italics. For example, MEKK1, GH3, NDK/NME, whose expression changes were discussed in Section 3.8. The text should also be checked for such errors.
4. In Section 2.1, it is highly desirable to indicate at what time of day each sample was collected, if such data are preserved. Circadian rhythms can strongly influence transcriptomes. It is also desirable to indicate the coordinates at which the biomaterial was collected.
5. What is written in Section 2.2 should probably be moved to the Results section, as it describes what was identified rather than the identification process itself. There are also some notes about the results obtained. Can the pathogen be identified more accurately than to the family level (e.g., by ITS-region sequencing)? According to NCBI Taxonomy, Melampsora is the only genus in the family Melampsoraceae, and the family itself belongs to the phylum Basidiomycota, subphylum Pucciniomycotina, class Pucciniomycetes, order Pucciniales. And I suppose it is better to use modern names of taxa.
6. In Section 2.4, it is worth describing in as much detail as possible all the programmes used in the analysis, including their versions. For example, fastqc for assessing the quality of reids, STAR/HISAT2/kallisto for mapping them, DESeq2/edgeR for finding differentially expressed genes, etc. The reference gene or transcriptome used should also be specified.
7. Section 2.5 should detail all reagents and equipment used, including their manufacturer. A similar section should describe the process of RNA isolation, library preparation, and sequencing.
8. It is highly desirable to specify how the raw sequencing data from the study can be accessed. For example, to provide the accession number in the SRA repository if the data are hosted there.
9. The captions to Figures 2, 3, 4, 5 should be made more detailed, indicating by what methods each part of the image was obtained.
10. Figures 6, 7, 8, 9 should be made larger and in higher resolution. Currently, it is necessary to zoom in on the document or strain vision to see signatures, especially in Figure 7.
11. The left volcanic plot has a region of points that behave atypically for volcanic plots, being far away from log2(FC)=0 with high density. Is it possible to explain what this behaviour is related to? Could it be interesting in terms of process biology or is it an artefact of the study? In the file attached below I have highlighted them with a blue square.
12. The explanation for the differential expression of hormone-related genes may differ from what is written in the article. In very brief, plants have 2 blocks of immune response regulation: an SA-regulated response against biotrophic pathogens (as noted in the article, it is associated with systemic acquired resistance and hypersensitive response) and a JA/ET module active against necrotrophs. Moreover, in Arabidopsis, wheat and rice, they are in antagonistic relationships and inhibit each other (since, for example, reactive oxygen species may be useful against biotrophs but, conversely, will help necrotrophic pathogens) (https://doi.org/10.3390/ijms20030671). For trees such as poplar, current evidence refutes this - perhaps these hormones act synergistically (https://doi.org/10.1111/nph.18148). Melampsora is a well-known example of a biotrophic pathogen. So perhaps suppression of JA signalling is linked to suppression by SA. I would advise the authors to analyse the expression of as many genes related to SA signalling as possible (all PRs, NRP1, 3, 4, defensin, etc.) as well as JA and ET. Perhaps the fungus is indeed suppressing the plant's immune response.
13. The data obtained in this study are in any case very valuable. But I would ask the authors to specify more clearly why the L group was analysed. As far as I understood, it is related to the fact that in the second year plants are less susceptible to rust due to a denser and differently structured cover. But it would be better to state this more explicitly, preferably both in the abstract and in the introduction.

Author Response
For research article
|
Response to Reviewer 1 Comments
|
||
|
1. Summary |
|
|
|
Thank you very much for taking the time to review this manuscript. Please find the detailed responses below and the corresponding revisions/corrections highlighted/in track changes in the re-submitted files. |
||
|
2. Questions for General Evaluation |
Reviewer’s Evaluation |
Response and Revisions |
|
Does the introduction provide sufficient background and include all relevant references? |
Yes/Can be improved/Must be improved/Not applicable |
Yes |
|
Are all the cited references relevant to the research? |
Yes/Can be improved/Must be improved/Not applicable |
Yes |
|
Is the research design appropriate? |
Yes/Can be improved/Must be improved/Not applicable |
Yes |
|
Are the methods adequately described? |
Yes/Can be improved/Must be improved/Not applicable |
Yes |
|
Are the results clearly presented? |
Yes/Can be improved/Must be improved/Not applicable |
Yes |
|
Are the conclusions supported by the results? |
Yes/Can be improved/Must be improved/Not applicable |
Yes |
|
3. Point-by-point response to Comments and Suggestions for Authors |
||
|
Comments 1: [Paste the full reviewer comment here.] This is a mostly formal remark, but references in the text should be arranged in accordance with the requirements of the journal, i.e. their numbers should be written in bold coloured font in square brackets, it is better to clarify this on the site. Now there are only numbers in the text, and this should be corrected |
||
|
Response 1: Thank you for pointing this out. We agree with this comment. Therefore, We have the following changes: The full text reference numbers were written in the corresponding square brackets, as required by the journal, which has been corrected;this change can be found – page 1, in Introduction paragraph 1, and line3、line7、line11;page 2, paragraph 1, and line2, line3, line5, line7, line8;page 2, paragraph 2, and line4;line5、line8;page 2, paragraph 3, and line4、line9;page 2, paragraph 2, and line4, line5, line8;page 2, paragraph 3, line4, line9; page 2, paragraph 4,line2, line3, line8, line11, line12,line14, line17;page 4, in Section 2.2 and line6; page 4, in Section 2.4 and line3, line7, line11;page 6,in Section 3.1 and line11; page 15, in Section 4. paragraph 2, and line3, line5; page 16, paragraph 1 line2, line4, line6, line8, line13; page 16, paragraph 2 line4, line5, line8, line13; page 16, paragraph 3 line2, line5; page 16, paragraph 4 line3, line5, line11 ; page 17, paragraph 1 line5; page 17, paragraph 2 line4, line7, line13, line14; page 17, paragraph 3 line2, line4; page 17, paragraph 4 line2, line5, line7, line9, line11 ; page 17, paragraph 5 line3, line7, line9 ; page 18, paragraph 1 line1, line6 “[updated text in the manuscript if necessary]” |
||
|
Comments 2: [Paste the full comment here.] It is also a formal point, but the names of species and genera should always be written in italics. It is necessary to make amends to“Phytophthora infestans” (line 40), “Populus euphratica” (line 52), “Pinus massoniana” (line 63), “Picea cracifolia” (line 83 and line 99, line 390) and possibly something else. Please check the manuscript again. |
||
|
Response 2: Agree. Thank you for pointing this out. We agree with this comment. Therefore, We have the following changes: According to the revision of Jof-2906375 review2, we have deleted the reference to Phytophthora infestans and Pinus massoniana content; page 2, paragraph 4 line13 Replace "Populus euphratica" with "Populus euphratica"; Replaced all occurrences of "Picea cracifolia" in the article with "Picea cracifolia" page 1, paragraph 1 line2, line18, line20; page 1, paragraph 2 line1, line4, line7,line10; page 2, paragraph 4 line17, line20, line22; page 6,in Section 3.1 line2, line5; page 15,Section 4, paragraph 1 line1 line3, line5; page 17, paragraph 1 line1; page 18, in Section5 paragraph 1 line2 paragraph 4 line2; page 19 References3, 4, 7 “[updated text in the manuscript if necessary]” |
||
|
Comments 3: [Paste the full reviewer comment here.] Similarly, when referring to genes rather than proteins, they should be written in italics. For example, MEKK1, GH3, NDK/NME, whose expression changes were discussed in Section 3.8. The text should also be checked for such errors. |
||
|
Response 3: Agree. Thank you for pointing this out. We agree with this comment. Therefore, We have the following changes: in the revised manuscript this change can be found – page2, paragraph3 line6,Replace " MAPK " with "MAPK"; Replace " MAPKs " with "MAPKs";in the revised manuscript this change can be found – in Section 3.8.page 15,line3, line4, line5, line6, line7, line8, line9, line10; page 16,in Section4 paragraph1 line14; page 17, paragraph 3 line1, line6, line2, line3, line5; page 17, paragraph 5 line3, line4, line5, line6, line7, line9; page 18, paragraph 2 line1, line4; page 18, in Section5 paragraph 2 line4, line5; page 18, paragraph 3 line6, line12 “[updated text in the manuscript if necessary]” |
||
|
Comments 4: [Paste the full reviewer comment here.] In Section 2.1, it is highly desirable to indicate at what time of day each sample was collected, if such data are preserved. Circadian rhythms can strongly influence transcriptomes. It is also desirable to indicate the coordinates at which the biomaterial was collected. |
||
|
Response 4: Agree. Thank you for pointing this out. We agree with this comment. Therefore, We have the following changes: in the revised manuscript this change can be found – in Section 2.1 “[updated text in the manuscript if necessary]” |
||
|
Comments 5: [Paste the full reviewer comment here.] What is written in Section 2.2 should probably be moved to the Results section, as it describes what was identified rather than the identification process itself. There are also some notes about the results obtained. Can the pathogen be identified more accurately than to the family level (e.g., by ITS-region sequencing)? According to NCBI Taxonomy, Melampsora is the only genus in the family Melampsoraceae, and the family itself belongs to the phylum Basidiomycota, subphylum Pucciniomycotina, class Pucciniomycetes, order Pucciniales. And I suppose it is better to use modern names of taxa. |
||
|
Response 5: Agree.Thank you for pointing this out. We agree with this comment. Therefore, We have the following changes: Section 2.2 has been moved to the results section and categorized using modern names;in the revised manuscript this change can be found – in Section 3.1 “[updated text in the manuscript if necessary]” |
||
|
Comments 6: [Paste the full reviewer comment here.] In Section 2.4, it is worth describing in as much detail as possible all the programmes used in the analysis, including their versions. For example, fastqc for assessing the quality of reids, STAR/HISAT2/kallisto for mapping them, DESeq2/edgeR for finding differentially expressed genes, etc. The reference gene or transcriptome used should also be specified. |
||
|
Response 6: Agree. Thank you for pointing this out. We agree with this comment. Therefore, We have the following changes: in the revised manuscript this change can be found – in Section 2.2, 2.4 “[updated text in the manuscript if necessary]” |
||
|
Comments 7: [Paste the full reviewer comment here.] Section 2.5 should detail all reagents and equipment used, including their manufacturer. A similar section should describe the process of RNA isolation, library preparation, and sequencing. |
||
|
Response 7: Agree. Thank you for pointing this out. We agree with this comment. Therefore, We have the following changes: Discuss the changes made, providing the necessary explanation/clarification. Mention exactly where in the revised manuscript this change can be found – page number, paragraph, and line.in the revised manuscript this change can be found – in Section 2.3, 2.5, Table 1, Table 2,Table 3 “[updated text in the manuscript if necessary]” |
||
|
Comments 8: [Paste the full reviewer comment here.] It is highly desirable to specify how the raw sequencing data from the study can be accessed. For example, to provide the accession number in the SRA repository if the data are hosted there. |
||
|
Response8: Agree. Due to the large amount of data and cumbersome operation, the login number will be sent to you later. “[updated text in the manuscript if necessary]” |
||
|
Comments 9: [Paste the full reviewer comment here.] The captions to Figures 2, 3, 4, 5 should be made more detailed, indicating by what methods each part of the image was obtained. |
||
|
Response 9: Agree. Thank you for pointing this out. We agree with this comment. Therefore, We have the following changes: in the revised manuscript this change can be found – in Figure 2, Figure 3, Figure 4 “[updated text in the manuscript if necessary]” |
||
|
Comments 10: [Paste the full reviewer comment here.] Figures 6, 7, 8, 9 should be made larger and in higher resolution. Currently, it is necessary to zoom in on the document or strain vision to see signatures, especially in Figure 7. |
||
|
Response 10: Agree. Thank you for pointing this out. We agree with this comment. Therefore, We have the following changes: in the revised manuscript this change can be found – in Figure 6, 7, 8, 9 “[updated text in the manuscript if necessary]” |
||
|
Comments 11: [Paste the full reviewer comment here.] The left volcanic plot has a region of points that behave atypically for volcanic plots, being far away from log2(FC)=0 with high density. Is it possible to explain what this behaviour is related to? Could it be interesting in terms of process biology or is it an artefact of the study? In the file attached below I have highlighted them with a blue square. |
||
|
Response11: Thank you for pointing this out. First, this figure is a volcano plot of healthy one-year-old needlesand diseased one-year-old needles; the red dots indicate differentially up-regulated genes, and the high-density regions away from log2(FC)=0 are some of the more differentially up-regulated genes “[updated text in the manuscript if necessary]” |
||
|
Comments 12: [Paste the full reviewer comment here.] The explanation for the differential expression of hormone-related genes may differ from what is written in the article. In very brief, plants have 2 blocks of immune response regulation: an SA-regulated response against biotrophic pathogens (as noted in the article, it is associated with systemic acquired resistance and hypersensitive response) and a JA/ET module active against necrotrophs. Moreover, in Arabidopsis, wheat and rice, they are in antagonistic relationships and inhibit each other (since, for example, reactive oxygen species may be useful against biotrophs but, conversely, will help necrotrophic pathogens) (https://doi.org/10.3390/ijms20030671). For trees such as poplar, current evidence refutes this - perhaps these hormones act synergistically (https://doi.org/10.1111/nph.18148). Melampsora is a well-known example of a biotrophic pathogen. So perhaps suppression of JA signalling is linked to suppression by SA. I would advise the authors to analyse the expression of as many genes related to SA signalling as possible (all PRs, NRP1, 3, 4, defensin, etc.) as well as JA and ET. Perhaps the fungus is indeed suppressing the plant's immune response. |
||
|
Response 12: Agree. Thank you for pointing this out. We agree with this comment. Therefore, We have the following changes: in the revised manuscript this change can be found – page 16, paragraph 2, 3 “[updated text in the manuscript if necessary]” |
||
|
Comments 13: [Paste the full reviewer comment here.] The data obtained in this study are in any case very valuable. But I would ask the authors to specify more clearly why the L group was analysed. As far as I understood, it is related to the fact that in the second year plants are less susceptible to rust due to a denser and differently structured cover. But it would be better to state this more explicitly, preferably both in the abstract and in the introduction. |
||
|
Response 13: Agree. Thank you for pointing this out. We agree with this comment. Therefore, We have the following changes: in the revised manuscript this change can be found – page 1, Abstract and Introduction “[updated text in the manuscript if necessary]” |
||

Reviewer 2 Report
The article is interesting but the writting of the document and the explanation of the data deserve more time of editing.
The article should be improved greatly

Author Response
For research article
|
Response to Reviewer X Comments
|
||
|
1. Summary |
|
|
|
Thank you very much for taking the time to review this manuscript. Please find the detailed responses below and the corresponding revisions/corrections highlighted/in track changes in the re-submitted files. |
||
|
2. Questions for General Evaluation |
Reviewer’s Evaluation |
Response and Revisions |
|
Does the introduction provide sufficient background and include all relevant references? |
Yes/Can be improved/Must be improved/Not applicable |
Yes |
|
Are all the cited references relevant to the research? |
Yes/Can be improved/Must be improved/Not applicable |
Yes |
|
Is the research design appropriate? |
Yes/Can be improved/Must be improved/Not applicable |
Yes |
|
Are the methods adequately described? |
Yes/Can be improved/Must be improved/Not applicable |
Yes |
|
Are the results clearly presented? |
Yes/Can be improved/Must be improved/Not applicable |
Yes |
|
Are the conclusions supported by the results? |
Yes/Can be improved/Must be improved/Not applicable |
Yes |
Changes to the jof -2906375-review2 article
|
Comments 1: |
|
Response 1: Thank you for pointing this out. We agree with this comment. Therefore, We have the following changes: in the revised manuscript this change can be found – page 1, in Abstract and line 3, Replace "rust fungi " with "rust ". “[updated text in the manuscript if necessary]” |
|
Comments 2: [Paste the full reviewer comment here.] |
|
Response 2: Agree.Thank you for pointing this out. We agree with this comment. Therefore, We have the following changes: in the revised manuscript this change can be found – page 1, in Abstract and line 4, Replace "using the 7 Illumina sequencing platform for transcriptome analysis. " with "Using RNA-seq on the Illumina HiSeq500 platform, we analyzed" “[updated text in the manuscript if necessary]” |
|
Comments 3: [Paste the full reviewer comment here.] |
|
Response 3: Thank you for pointing this out. We agree with this comment. Therefore, We have the following changes: For simplicity, needles diseased one-year-old needles are replaced in this paper with B; healthy one-year-old needles are replaced in this paper with N; healthy two-year-old needles are replaced in this paper with L ;in the revised manuscript this change can be found – page 1, in Abstract and line 5, line 6;page 2, in Introduction paragraph4 and line 19, line 20;page 3 in Section 2.1, line 3,line 4;page 9 in Section 3.4, line 2,line 3;page 10 in Section 3.5,line 2,line 4;page 11 in Section 3.6,line 3,page 13 in Section 3.7, line 1,line 5; page 15 in Section 3.7, page 15, in Section 4 Discussion, paragraph 2, and line1, page 17, paragraph 2, and line1, page 18, in Section 5 Conclusions, paragraph 2, and line1,paragraph 3, and line2. “[updated text in the manuscript if necessary]” |
|
Comments 4: [Paste the full reviewer comment here.] |
|
Response 4: Agree. Thank you for pointing this out. We agree with this comment. Therefore, We have the following changes: in the revised manuscript this change can be found – page 1, in Abstract and line 6 “[updated text in the manuscript if necessary]” |
|
Comments 5: [Paste the full reviewer comment here.] Gene Ontology (GO) annotation and Kyoto Ency- 11 clopedia of Genes and Genomes (KEGG) enrichment analyses were conducted. In N vs B, significant 12 enrichment of DEGs was observed in metabolic and cellular processes, |
|
Response 5: Agree. Thank you for pointing this out. We agree with this comment. Therefore, We have the following changes: This section has been modified due to a change in the content of the article summary,in the revised manuscript this change can be found – page 1, in Abstract and line 7, line8. “[updated text in the manuscript if necessary]” |
|
Comments 6: [Paste the full reviewer comment here.] |
|
Response 6: Agree. Thank you for pointing this out. We agree with this comment. Therefore, We have the following changes: in the revised manuscript this change can be found – page 1, in Introduction paragraph1, and line7-9 “[updated text in the manuscript if necessary]” |
|
Comments 7: [Paste the full reviewer comment here.] |
|
Response 7: Agree. Thank you for pointing this out. We agree with this comment. Therefore, We have the following changes: in the revised manuscript this change can be found –page 1, in Introduction paragraph2, and line1-8 “[updated text in the manuscript if necessary]” |
|
Comments 8: [Paste the full reviewer comment here.] |
|
Response 8: Agree. Thank you for pointing this out. We agree with this comment. Therefore, We have the following changes: in the revised manuscript this change can be found – page2, paragraph4, and line5-14. “[updated text in the manuscript if necessary]” |
|
Comments 9: [Paste the full reviewer comment here.] |
|
Response 9: Agree. Thank you for pointing this out. We agree with this comment. Therefore, We have the following changes: in the revised manuscript this change can be found – page2, paragraph4, and line5 “[updated text in the manuscript if necessary]” |
|
Comments 10: [Paste the full reviewer comment here.] |
|
Response 10: Agree. Thank you for pointing this out. We agree with this comment. Therefore, We have the following changes: in the revised manuscript this change can be found – page2, paragraph4, and line1-23 “[updated text in the manuscript if necessary]” |
|
Comments 11: [Paste the full reviewer comment here.] |
|
Response 11: Agree. Thank you for pointing this out. We agree with this comment. Therefore, We have the following changes: in the revised manuscript this change can be found – page2, in Section 2.1,line1-8. “[updated text in the manuscript if necessary]” |
|
Comments 12: [Paste the full reviewer comment here.] |
|
Response 12: Agree. Thank you for pointing this out. We agree with this comment. Therefore, We have the following changes: in the revised manuscript this change can be found – page4, in Section 2.5,line1-8. “[updated text in the manuscript if necessary]” |
|
Comments 13: [Paste the full reviewer comment here.] |
|
Response 13: Thank you for pointing this out. We agree with this comment. Therefore, We have the following changes: The purpose of 3.1 and 3.2 is to explore whether there is any significant difference in morphology between annual and biennial needles, and combined with the morphological observations, we will then mine the relevant differential genes from the molecular point of view. “[updated text in the manuscript if necessary]” |
|
Comments 14: [Paste the full reviewer comment here.] |
|
Response 14: Thank you for pointing this out. We agree with this comment. Therefore, We have the following changes: in the revised manuscript this change can be found – page10, in Section 3.5. “[updated text in the manuscript if necessary]” |
|
Comments 15: [Paste the full reviewer comment here.] |
|
Response 15: Thank you for pointing this out. We agree with this comment. Therefore, We have the following changes: in the revised manuscript this change can be found – page7, in Section 3.2.line8. “[updated text in the manuscript if necessary]” |
|
Comments 16: [Paste the full reviewer comment here.] |
|
Response 16: Agree. Thank you for pointing this out. We agree with this comment. Therefore, We have the following changes: This part has been removed from the article “[updated text in the manuscript if necessary]” |
|
Comments 17: [Paste the full reviewer comment here.] |
|
Response 17: Agree. Thank you for pointing this out. We agree with this comment. Therefore, We have the following changes: in the revised manuscript this change can be found – page15, in Section 3.8.line3. “[updated text in the manuscript if necessary]” |

Round 2
Reviewer 1 Report
The authors have substantially improved the manuscript by responding consistently to all comments. I have no further significant comments and suggest to accept the revised manuscript for publication.
I have no further significant comments.
Reviewer 2 Report
This study has yielded highly positive conclusions. Transcriptome sequencing of both healthy and diseased samples of Picea crassifolia identified differentially expressed genes (DEGs), confirmed through RT-qPCR expression analysis. A significant enrichment of DEGs was found in critical metabolic pathways in diseased samples, including photosynthesis, plant signaling, interactions with pathogens, and phenylalanine synthesis. Moreover, morphological differences were observed in one-year-old and two-year-old needles, with the latter exhibiting reduced stomatal density, increased lignin content, and a thicker wax layer.
Enrichment analysis revealed higher gene activity related to stomatal development and wax synthesis in older needles. These findings advance our understanding of Picea crassifolia's genetic responses to rust infection, laying the groundwork for future research in plant pathology and genetics, particularly in disease resistance mechanisms in conifer species.